# The Intra- and Extra-Telomeric Role of TRF2 in the DNA Damage Response

**DOI:** 10.3390/ijms22189900

**Published:** 2021-09-14

**Authors:** Siti A. M. Imran, Muhammad Dain Yazid, Wei Cui, Yogeswaran Lokanathan

**Affiliations:** 1Centre for Tissue Engineering and Regenerative Medicine, Faculty of Medicine, Universiti Kebangsaan Malaysia, Jalan Yaacob Latiff, Bandar Tun Razak, Cheras, Kuala Lumpur 56000, Malaysia; siti.imran@ukm.edu.my (S.A.M.I.); dain@ukm.edu.my (M.D.Y.); 2Institute of Reproductive and Developmental Biology, Faculty of Medicine, Imperial College London, Du Cane Road, London W12 0NN, UK; wei.cui@imperial.ac.uk

**Keywords:** TRF2, DNA damage response, telomeres protection, extra-telomeric

## Abstract

Telomere repeat binding factor 2 (TRF2) has a well-known function at the telomeres, which acts to protect the telomere end from being recognized as a DNA break or from unwanted recombination. This protection mechanism prevents DNA instability from mutation and subsequent severe diseases caused by the changes in DNA, such as cancer. Since TRF2 actively inhibits the DNA damage response factors from recognizing the telomere end as a DNA break, many more studies have also shown its interactions outside of the telomeres. However, very little has been discovered on the mechanisms involved in these interactions. This review aims to discuss the known function of TRF2 and its interaction with the DNA damage response (DDR) factors at both telomeric and non-telomeric regions. In this review, we will summarize recent progress and findings on the interactions between TRF2 and DDR factors at telomeres and outside of telomeres.

## 1. Introduction

Telomeres are a specific DNA-protein structure at the end part of each chromosome, containing both telomeric DNA repeats TTAGGG and their associated shelterin protein complex. Telomeric DNA requires this protein complex to assist in maintaining its length and to protect it from becoming shortened through DNA synthesis [1]. The shelterin complex is composed of six members: the telomere repeat binding factors 1 and 2 (TRF1 and TRF2) [2], repressor/activator protein 1 (Rap1) [3,4], TRF1- and TRF2-interacting nuclear protein 2 (TIN2) [5], protection of the telomeres 1 (POT1) [6,7] and *ACD* gene (TPP1) [8,9]. In addition to its role in forming the t-loop structure [10,11], TRF2 has also been reported to actively inhibit the ataxia telangiectasia-mutated (ATM) kinase from recognizing the telomere end as a DNA break [12]. This protective role of TRF2 is beneficial at the telomeres and in the survival of tumor cells, where the latter would be detrimental for normal biological function and mortality.

The function that TRF2 plays at the telomeres is crucial in maintaining the proper telomere length that will prevent early aging or the development of age-related diseases [13]. Furthermore, it has an important role in regulating DNA damage response, which will prevent chromosomal instability, hence, the development of more severe diseases, such as cancer [14]. Previous studies have mainly focused on interaction of TRF2 with DNA damage response (DDR) factors at the telomeres. However, the interaction between TRF2 and DDR factors has been shown to be broader than the scope of telomeres, as studies have found the colocalization of TRF2 with other DDR factors at the site of DNA breaks of other chromosomal regions [15,16]. As TRF2 mainly plays an inhibitory role towards the DDR at the telomeres, it may have a stimulatory effect in DNA damage repair as TRF2 was found to be recruited at the non-telomeric DNA damage site along with other DNA damage ‘sensors’ [16]. The double-edged sword effect that TRF2 has would be highly interesting to discuss and should be further explored for its specific functions that may have possible therapeutic implications.

## 2. DNA Damage Response (DDR)

DNA damage could occur in cells either endogenously, through normal cellular replication [17,18] and metabolism [19,20], or exogenously through ultraviolet (UV) [21,22], ionizing radiation (IR) [23,24] or various genotoxic compounds [25,26] that could induce DNA damage. Different stressors will cause different types of DNA damage. Normal DNA replication could induce mismatch of the nucleotide and cause mutations [27,28,29,30]. Stressors such as oxidative stress will produce reactive oxygen species (ROS) from normal cellular metabolism or from external genotoxic compound [31,32,33], which will cause DNA breaks, either single-stranded or double-stranded [34,35,36]. Unrepaired DNA damage could cause severe mutations and chromosomal instability, which would have detrimental effects on the cells and lead to cell death, while DNA breaks that are repaired through non-homologous end joining (NHEJ) might cause mutations during the process.

The DDR is the response mechanism which will detect any DNA damage that occurs throughout the chromosome and will activate a repair cascade to the damage site. This will help the cells either to proliferate normally if the repair was successful or to activate the cellular programmed cell death if the damage was too extensive and was unable to be repaired. The known DNA damage repair mechanisms include mismatch repair (MMR) [37], base excision repair (BER) [38], nucleotide excision repair (NER) [39], homologous recombination (HR) [40] and non-homologous end joining (NHEJ) [41]. Specific types of DNA damage could be fixed by a specific repair factor, such as the ATM kinase, which is the main factor in double-strand break repair through NHEJ [42]. Figure 1 shows the causes and types of DNA damage as well as the response cascade involved in repairing the damages.

DNA damage will activate a series of sensors, transducers and effectors in coordinating and regulating the DDR mechanism [43,44]. DNA damage that occurs at the chromosome is primarily coordinated by the phosphatidylinositol-3-kinase-related protein kinase (PIKK) family [45,46]. The primary signaling kinases of the PIKK family that are involved in the DDR are the DNA-dependent protein kinase catalytic subunit (DNA PKcs), ATM and ATM and Rad3-related kinase (ATR) [47], mammalian target of rapamycin (mTOR) [48,49] and suppressor with morphological effect on genitalia family member (SMG1) [50].

The relationship between DNA damage and TRF2 is complicated as DNA damage could either upregulate or downregulate TRF2 expression [51,52], whereas the disruption in the telomere integrity and function, such as mutation of TRF2, has been shown to increase DDR [53]. The occurrence of DNA damage has been reported to transiently increase TRF2 expression [51], while accumulation of non-telomeric DNA damage would further cause degradation of TRF and telomere shortening, which will further lead to DNA instability [52]. The changes of TRF2 expression with DNA damage are also reflected in cancers. Elevated TRF2 expression has been identified as frequently occurring during transformation of breast tumor cells as well as in colorectal cancers, which promote tumor formation and progression [54,55,56]. On the other hand, downregulation of TRF2 has been found in Hodgkin’s lymphoma [57,58]. This loss of TRF2 in Hodgkin’s lymphoma disrupts the telomere-TRF2 interaction, leading to chromosomal rearrangement [59]. The unique role of TRF2 is even more puzzling when an increased expression of TRF2 would also induce DNA damage, which is mostly found in some cancer cells [54,55,60,61,62,63]. The overexpression of TRF2 was found to cause replication stalling at the telomeres; hence, it causes telomere attrition [64].

Over the years, extensive research has been conducted to investigate the connection between telomere protection and the DDR, where a number of interconnections between the factors have been established. It has been reported that TRF2 expression can be regulated at both the transcriptional and post-transcriptional levels depending on the cell types and other factors [51], as it has been shown that TRF2 mRNA level is similar [56] but the protein level is different in different types of cells [65]. However, the regulation of TRF2 levels may also be regulated by post-translational modification, such as phosphorylation, which may affect the stability of TRF2 protein.

## 3. TRF2 Post-Translational Modification and Its Involvement with DDR

There are not many reports on the post-transcriptional modifications (PTM) of TRF2 that occur to modulate its function in the biological processes. To date, the modifications of TRF2 are known to assist in regulating its stability, binding activity and localization, either at the telomeres or outside of the telomeres [66]. Several PTM sites have been found at different domains of TRF2 as shown in the PhosphoSites database [67], where the phosphorylation event has taken up the most. More than 150 entries, including research findings and technical notes, have reported on Ser365 phosphorylation of TRF2, indicating that this phosphorylation may have an important role in regulating TRF2 functions. It has been demonstrated that phosphorylation of TRF2 at Ser365 by CDK [68] coordinates the assembly and disassembly of t-loops during the cell cycle, which protects telomeres from replication stress and an unscheduled DNA damage response [69]. In another study, phosphorylation of TRF2 on serine 323 by extracellular signal-regulated kinase (ERK1/2) was found in both normal and cancer cells [70]. Furthermore, human TRF2 has been reported to contain two highly conserved PIKK phosphorylation sites at Thr188 and Ser368, which can phosphorylate ATM upon induction of DNA damage [71,72]. The phosphorylation at Thr188 occurred 30 min post irradiation along with ATM-dependent factors, namely pp53 (ser15) and γH2AX, then gradually decreased and completely disappeared by 2 h [72]. Other downstream factors that will be activated by ATM kinase in response to radiation-induced DNA damage include Chk2 that will induce further DDR cascade [73]. Chk2 has been reported to phosphorylate TRF2 at Ser20, which will decrease its binding affinity at the telomeres [74,75]. This phosphorylation of TRF2 was thought to be important in the relocalization of TRF2 to non-telomeric DSBs. These findings suggest that phosphorylation of TRF2 may have a function in DNA repair.

SUMOylation is the attachment of a small conjugating enzyme (ubiquitin-like modifier, SUMO) to lysine residue of target proteins [76,77]. SUMOylation of TRF2 helps to stabilize it and assists in its binding to the telomeres [78] as well as protecting the telomeres from NHEJ [3]. MMS21 SUMO ligase of the SMC5/6 complex SUMOylates TRF2, which is essential for the recruitment of TRF2 to the telomeres, along with other shelterin subunits such as TRF1 and Rap1 [79,80]. However, SUMOylation of TRF2 was found to occur at dysfunctional telomeres where it causes alternative lengthening of the telomeres (ALT) [81]. Such modification may also cause the uncontrollable lengthening of the telomere length that is observed in some cancer cells [79]. TRF2 has also been shown to be SUMOylated by PIAS1 that preferred the telomere-unbound form, where the telomere-dissociated TRF2 will be ubiquitinated by RNF4 and degraded by a proteasome [78]. However, little is known of SUMOylation function in the mammalian cells, whereas the SUMOylation mechanism of TRF2 is still unknown.

RNF4 is a mammalian SUMO-targeted ubiquitin ligase that contains SIM and RING domains, which interacts with several SUMO-conjugated proteins to promote ubiquitination [82]. Ubiquitination targets protein for proteasome degradation. RNF4 acts together with PIAS1 in regulating the levels of TRF2 at the telomere, which is important in maintaining the proper function of TRF2 as the telomere protection [78,83]. RNF4 will disrupt the stability of SUMOylated TRF2 but not the unmodified TRF2 [78]. RNF4 has also been shown to be important in the DDR either at the telomeres or outside of the telomeres where it was found to be colocalized with other DDR factors and was found to have a function in the downstream of the ATM-kinase pathway [84]. Another essential ubiquitination of TRF2 is by Siah1, where the latter is activated by ATM in response to DNA damage [81]. The relationship between Siah1 and TRF2 also works as a positive feedback loop with p53 in regulating DDR and initiating DNA repair, where p53 was shown to induce Siah1, which represses TRF2, where low levels of TRF2 will increase activation of p53 via ATM kinase pathway [85]. Another recent finding of TRF2 PTM was its acetylation by P300 at lysine residue 293, which was thought to be important in TRF2 stabilization and binding efficiency at the telomeres [86]. P300 has been found to be essential in the activation and regulation of DDR as it acetylates various factors involved in DNA damage repair [87]. P300 was also found to acetylate another PTM enzyme, Poly(ADP-ribosyl)ation or PARsylation 1 (PARP1), which is one of the earliest responses of DDR in identifying the types of damage [88,89]. PARP1 will recruit other DDR factors at the site of DNA damage that will initiate the repair cascade [90,91]. PARP1 was further identified with Poly(ADP-ribosyl)ate TRF2, which will affect the binding of TRF2 at the telomeres [92]. However, this effect has not been shown at other chromosomal regions.

It has been reported that TRF2 recruited other repair factors at the damage site and acted as the early response to DDR, similar to p53 binding protein 1 (53BP1) and ATM [15,16,93]. However, TRF2 does not have the ability to repair DNA damage on its own; rather, it does so through interactions with multiple repair factors. Nonetheless, it plays an essential role in regulating and maintaining gene stability. These PTMs of TRF2 are thought to be the key in exploring the mechanisms involving TRF2 and DDR at other non-telomeric sites. PTM of TRF2 has been shown to be important in its involvement with DDR; however, it was only found to be specific at the telomeres [80,94]. Table 1 summarizes the PTMs of TRF2 by various modification enzymes.

## 4. TRF2 Protects Telomeres by Blocking the Activation of DDR

TRF2 is composed of four distinct domains: (1) the Myb domain (similar to TRF1), which confers specificity for TTAGGG repeats of the telomeres; (2) the N-terminal basic domain; (3) the TRF homology (TRFH) domain, which is involved in homodimerization with TRF1 and binding to accessory factors; and (4) the hinge domain, which mediates the interactions with Rap1 and TIN2. The main domains of TRF2 that bind to the DNA sequence are the N-terminal and C-terminal Myb domains [98]. TRF2, along with other shelterin complex subunits, primarily functions to protect the telomere end from being recognized as a DNA break [99,100] as well as from end-to-end fusions [101]. The binding of TRF2 via its binding domains at the telomeric sequence safeguards the t-loop structure of telomeres from homologous recombination and protects the telomere end from being recognized as a DNA break [102].

As mentioned above, TRF2 has been reported as actively inhibiting the DDR pathway, namely the ATM kinase, from recognizing the telomere as a DNA break, thus protecting the genomic integrity [12]. During DNA replication, the ‘closed’ state of the telomeres will be unfolded to become the ‘open’ state, in which the t-loop unfolds and exposes the 3′-overhang. These uncapped telomeres could be recognized as a DSB and detected by the DDR pathway factors, thus leading to unwanted end-to-end fusion of the telomere end [72,103]. TRF2 inhibition of DDR has been shown to be important in maintaining telomere length and stability, which could lead to cancer or other age-related diseases [104].

The shelterin complex, specifically TRF2 and POT1, can inhibit the ATM and ATR kinases, respectively, from recognizing the telomeres as a break, where ATM recognition at the telomeres was shown in TRF2-deficient cells [105,106] and the repression of ATR was elevated after POT1 deletion [106]. TRF2 and POT1 inhibit the ATM and ATR independently. Furthermore, TRF2 protection at the telomeres is only through the inhibition of the ATM and not the ATR, which is evidenced by the fact that deletion of TRF2 did not induce any activation of Chk1, which is downstream of the ATR pathway [75,107] but affected the activation of the downstream factors of ATM pathway [108]. Moreover, ATR inhibition was shown to have no effect on the telomeres-dysfunction induced foci (TIF) formation [106], a phenomenon occurring when the chromosome ends become too short or become uncapped and are prone to be recognized as a DNA break by the DDR [53,109]. Nonetheless, TIF formation recruited 53BP1, which was found to be induced in the absence of TRF2 [53].

The mechanism of ATM inhibition by TRF2 at the telomeres needs to be further explored. The next section will further discuss TRF2 inhibition on ATM kinase and its downstream factors during DDR.

## 5. TRF2 Inhibits the ATM Kinase

ATM is a 370-kDa protein with a carboxyl-terminal sequence homologous to the catalytic domain of phosphatidylinositol 3-kinases (PI3K) [110]. ATM deficiency causes the genetic disorder ataxia-telangiectasia (A-T), characterized by cerebellar degeneration, immunodeficiency, radiation sensitivity, chromosomal instability and cancer predisposition [111]. Activation of ATM requires a functional Mre11-Rad50-Nbs1 (MRN) complex for a timely activation of ATM-mediated pathways in response to DNA double-strand breaks [112]. At the telomere, inhibition of ATM is regulated by TRF2 protein [12]. Critical loss of the telomere repeats or the loss of TRF2 function will induce the activation of ATM through the autophosphorylation at S1981 [12], similar to the signaling cascade of a DSB repair at other chromosomal regions [113,114,115]. As ATM is required for the activation of DNA damage repair cascade, its disruption or inhibition would hinder normal biological development [116].

There are two ways by which TRF2 inhibits the ATM kinase. The first one is by actively inhibiting the autophosphorylation of ATM at S1981, as this phosphorylation will activate ATM to activate its downstream factors, such as γH2AX and p53, thus initiating the DNA damage repair cascade [113]. TRF2 also inhibits ATM activation through the formation of t-loop [117]. When t-loop is formed at the telomeres, the telomere end will not be able to be recognized as a DNA strand break by ATM and other DDR factors [117].

## 6. TRF2 Involved in Non-Homologous End Joining (NHEJ)

Non-homologous end joining (NHEJ) is the primary DSBs repair pathway that operates throughout the cell cycle in higher eukaryotes. In NHEJ, the DSB is first recognized by Ku 70/80 heterodimer that will recruit other factors such as the DNA PKcs to initiate the DNA repair [118]. TRF2 has been reported to be involved in protecting the telomere end from unwanted NHEJ by inhibiting the heterotetramerization of Ku70–Ku80 heterodimer. TRF2 interacts with the α-helix 5 domain of Ku70, which mediates Ku70–Ku80 heterotetramerization that is required in DNA repair. The Ku70–Ku80 heterodimer is an essential initiator component of c-NHEJ that is involved in telomere maintenance [119].

Recent studies have found that rapid phosphorylation of ATM is important in the NHEJ repair of DSBs [16,95]. Similar to the ATM, telomere overhang due to the loss of function of TRF2 also accumulates 53BP1, which will facilitate c-NHEJ-based repair [120]. A previous study had reported that TRF2 depletion will inhibit homologous recombination (HR) and delay Rad51 foci formation upon γ-irradiation but does not affect NHEJ, whereas the overexpression of TRF2 will stimulate HR and inhibit NHEJ [121]. Another study also showed that the formation of NHEJ and Holliday junction was found in cells with impaired or deficient TRF2 [11,122]. Uncapped telomeres due to TRF2 deficiency were shown to initiate NHEJ, especially at critically shortened telomeres [123]. However, the function of TRF2 in DNA repair via NHEJ has mainly been focused on the telomeres. Due to the close interaction (activation/inhibition) that TRF2 has with the factors involved in NHEJ, it may play a similar role at other non-telomeric regions.

It has been shown that TRF2 recruitment to the DSBs is dependent on the activation of other DNA damage sensors, such as PARP1, as well as the binding domains of TRF2 [16]. This recruitment was also shown not to be specific at the telomeres. A previous study had reported a direct interaction of TRF2 with the p38 MAPK upon the induction of DNA damage [124]. These reports suggest that there are many possibilities in which TRF2 may interact with the DDR, where TRF2 will further recruit other factors that will repair the damaged DNA. Nevertheless, the exact mechanism of TRF2 involvement in DNA repair is still unknown. Table 2 summarizes the interactions that TRF2 has with various DDR factors, either at the telomeres or at non-telomeric regions.

**Table 2 ijms-22-09900-t002:** DNA damage response factors that interact with TRF2 at the telomeric and non-telomeric regions.

Repair Mechanism	Gene	Known Function	Known Function	Location	Reference (s)
ATM Kinase	ATM	Initial sensing of double-strand breaks of DNA	TRF2 inhibits the activation (autophosphorylation) of ATM on S1981 at the telomeres	Telomeres and non-telomeric regions	[12,104,125]
	γH2AX	Recruitment and activation of other DNA repair factors	TRF2 localized at the same site upon DSB induction	Non-telomeric regions	[104,126]
Cell Cycle	pChk2	Cell cycle regulation	Inhibited by TRF2 via ATM kinase inhibition	Telomeres and non-telomeric regions	[102,127,128]
	MRE11	Part of the Mre11-Rad50-Nbs1 (MRN) complex to facilitate DNA damage repair	Mediates the maintenance of t-loop and DNA repair at the telomeres with TRF2/RAP1	Telomeres and non-telomeric regions	[129,130]
	Apollo (SNM1B)	Involved in the repair of DNA lesions	Localization at the telomeres is TRF2-dependent	Telomeres	[131,132]
			Direct interaction with endogenous TRF2		[133]
	Rap1	Regulation of telomere structure and integrity	Interacts with TRF2 to inhibit NHEJ	Telomeres	[3]
	Ku70/80	Initiates and is required for c-NHEJ and suppresses alt-NHEJ	Interacts with TRF2 to inhibit c-NHEJ	Telomeres and non-telomeric regions	[119,134,135]
	DNA-PK	DNA double-strand break repair complex	Inhibited by TRF2 to prevent c-NHEJ	Telomeres and non-telomeric regions	[136]
	53BP1	Early response to DNA double-strand breaks	Localization at DSB induced by inhibition of TRF2	Telomeres and non-telomeric regions	[53,120,137]
BER	pol β	Base-excision repair	Physically bound to TRF2 at the NH2-terminal and COOH-terminal domains	Telomeres and non-telomeric regions	[138,139,140]
	FEN1	Base-excision repair	Physically bound to TRF2 at the NH2-terminal and COOH-terminal domains	Telomeres and non-telomeric regions	[139,140]
	APE1	Base-excision repair	Indirectly stimulated by TRF2	Telomeres	[136]
	Lig1	Base-excision repair	Indirectly stimulated by TRF2	Telomeres	[140]
	PARP1	DNA nick sensor activated rapidly and transiently in response to DNA damage	Recruits TRF2 at DNA damage site	Non-telomeric regions	[92,97]
Others	LAP1	A lamina-associated protein that was recently found to be involved in DDR	Forms a complex with TRF2 upon DNA damage induction and binds at DNA damage site	Non-telomeric regions	[141]
	RTEL1	A helicase essential in DDR	Recruited by TRF2 at telomeres to promote t-loop unwinding	Telomeres and non-telomeric regions (not TRF2-dependent)	[142,143,144]

## 7. TRF2 Interacts with Factors of Base Excision Repair (BER)

Telomeric DNA is susceptible to oxidative damage by deamination, oxidation or alkylation mechanism which consequently causes base modification. Owing to this event, the base excision repair pathway will be activated to repair the damage by removing the damaged bases. BER pathway involves a series of proteins, namely DNA glycosylases, AP endonucleases, DNA polymerase, flap endonucleases and DNA ligase, where each works after another during base excision repair [38]. DNA polymerase beta (Pol β) and flap endonuclease 1 (FEN-1) were found to be physically bound to TRF2 at the NH2-terminal and COOH-terminal domains for DNA repair pathways at telomeres [139]. The binding of TRF2 to Pol β was shown to stimulate the latter and repair the damaged DNA [139]. The importance of Pol β in telomere protection was further shown when it was found to be recruited at the telomeres by TRF2 to assist in the telomeric synthesis by telomerase [139]. A disruption of Pol β was shown to alter the telomere length, which in turn will trigger a DNA repair cascade, despite the existence of TRF2, which shows the importance of Pol β in telomere length regulation [140]. The interaction of Pol β and TRF2 has also been shown to be outside of the telomeres where both were found to be colocalized at the non-telomeric DNA damage site; however, it may occur independently [138]. There are very few studies investigating the involvement of TRF2 in BER. The mechanism that may involve TRF2 and BER factors needs to be explored to further understand if there is any possible function that TRF2 has in the DNA repair cascade.

## 8. TRF2 Involvement in the DDR Outside of the Telomeres

The function of TRF2 in the DDR has been mainly investigated at the telomeres, while very little is known regarding its involvement in the DDR outside of the telomeres. Given the previously reported stimulatory and inhibitory effects of TRF2, it may have a more significant involvement in the DRR at other chromosomal regions that have a double-strand break. This was proven when TRF2 was found to be colocalized at the DNA damage site, together with other DDR factors, such as γH2AX [15]. Previous reports have shown that TRF2 is localized at the site of the laser-induced DNA break outside of the telomeres within seconds upon the DSB induction [95,122]. It has also been shown that TRF2 may bind to DNA irrespective of the sequence [145].

The recruitment of TRF2 at the DNA break site is transient, where within seconds, TRF2 was shown to localize at the site of DSB, but the expression gradually declines over time, and the recruitment is decreased after multiple exposures to induce DSBs [15,126]. This result suggests that there are other factors that are regulating the recruitment of TRF2 at the DSB site.

Initially, it was thought that one such factor is the phosphorylation of TRF2 by the ATM kinase [72]. However, phosphorylation and recruitment of TRF2 to the DSBs were also observed in ATM-deficient, AT-primary fibroblast cells, suggesting that ATM is not necessarily required for the phosphorylation and recruitment of TRF2 to the DSB site. A similar effect was again observed in DNA-PKcs-deficient, human glioma cell lines, which suggests that DNA-PKcs was also not essential in the TRF2 DSBs recruitment [15,146].

Another study had found that TRF2 transiently formed a complex with lamina-associated polypeptide 1 (LAP1) upon DNA break induction and the complex was recruited at the DNA damage site [141]. Further investigation is needed in order to address the mechanism of TRF2 recruitment to the DSB site and to explore the function of TRF2 in the repair of non-telomeric DNA DSB. Recent studies have found that rapid phosphorylation of ATM is important in the NHEJ repair of DSBs [16,95]. It has also been reported that TRF2 depletion will inhibit HR and delayed Rad51 foci formation upon γ-irradiation and overexpression of TRF2 will stimulate HR and inhibit NHEJ [93]. Nevertheless, the mechanism of TRF2 involvement in DNA repair is still unknown. It was also shown that TRF2 recruitment to the DSBs is dependent on the activation of another DNA damage sensor, PARP1, as well as the binding domains of TRF2 [16]. Another report also showed the direct interaction of TRF2 with the p38 MAPK upon the induction of DNA damage [124]. These reports show that there are many possibilities in which TRF2 may interact with the DDR, where TRF2 will further recruit other factors that will repair the damaged DNA. Figure 2 illustrates the factors in DNA damage response that may interact directly or indirectly with TRF, either at the telomeres or at other DNA damage sites.

## 9. Other Known Extra-Telomeric Functions of TRF2

Other than its role at the telomeres, TRF2 has also been found to localize at extra-telomeric regions of the chromosome [147,148,149]. The Myb-domain of TRF2 is highly specific to recognize and bind to the G-rich site of the telomeres [150]. Nonetheless, the other binding domain of TRF2, the N-terminal, plays central roles in recruiting TRF2 to non-telomeric regions, as TRF2 lacking in N-terminal was shown to be unable to bind to non-telomeric DSBs site [126,145,151]. The N-terminus of TRF2 was also shown to interact with Ku70 to assist in binding at the site of DNA break [134]. The binding of TRF2 at other chromosomal sites was shown to be structure-dependent (via G-quadruplex) and not sequence-dependent, which requires both the C-terminal binding domain and N-terminal basic domain [152]. G-quadruplex at the telomeres functions to protect and maintain the telomere length through the binding of telomerase during DNA synthesis and replication [153,154,155,156,157]. However, G-quadruplex at other chromosomal regions triggers DNA damage and causes DNA instability [158].

Binding of TRF2 outside of the telomeres has also been documented at the interstitial telomeric sequences (ITSs) throughout the genome [147]. This binding is stabilized by lamin A/C, factors that are essential for maintaining the chromosomal nuclear territories and forming a so-called interstitial t-loop (ITL) [159]. Lamin A/C is bound to telomeres and the expression of a dominant negative TRF2 mutant leads to a loss of telomere integrity as well as the formation of DNA damage foci at telomeres [160]. Impairment or loss of this 3D lamin A/C-TRF2 interaction at ITLs results in increased chromatin dynamics, alteration in gene expression, impaired chromosome stability and genomic integrity [161,162,163].

At the telomeres, TRF2 has been shown to be a protein hub that is able to either directly bind or indirectly affect and recruit other proteins to protect the telomeres [1,164,165]. However, the interaction of TRF2 with other factors outside of the telomeres as well as its function through such interaction has not been thoroughly explored. A previous study has shown that TRF2 binds within the promoter of the cyclin-dependent kinase CDKNIA (p21/CIP1/WAF1), indicating that TRF2 transcriptionally regulates p21 expression. This is through the engagement of the REST-CoREST-LSD1 repressor complex at the p21 promoter in a TRF2-dependent fashion [166].

A study by Ovando-Roche et al. (2015) showed that TRF2 is important in the neural development of human embryonic stem cells (hESCS), where overexpression level of TRF2 promoted the hESCs to differentiate into neural progenitors, whilst downregulating TRF2 hindered the cells from neural differentiation [167]. It was further demonstrated that TRF2 interacts with hREST in hESCs to promote neural differentiation [167,168] and to maintain neuronal characteristics [169,170]. This indicates that TRF2 interacts with multiple factors to regulate different mechanisms, not only at the telomeres, but also at other non-telomeric regions as well. The role of TRF2 as a protein hub showed a lot of potential functions that TRF2 may play outside of the telomeres.

More functions of TRF2 that are not specific to the telomeres have also been found, such as its role in regulating mitochondrial function through the interaction with mitochondrial sirtuin 3 gene (SIRT3) [171] and even immune responses that are important in targeting cancer cells [172,173]. This interrelation of TRF2 with various factors is important in regulating DNA replication as well as maintaining DNA stability. These interactions may also have significant importance in normal biological functions such as aging, where TRF2 may act with several factors involved in the development of age-related diseases [174]. The discovery of specific interactions that TRF2 has with a specific factor may act as a possible targeting potential in disease treatment. However, this may be a challenge, as TRF2 has both promoting and impeding functions in modulating various mechanisms at the molecular level. Finding the right balance in TRF2 level to have an optimal function would be essential.

## 10. Conclusions and Future Directions

The role of TRF2 has mainly been focused on the telomeres. However, it has been shown that TRF2 function is not specific to the telomeres. Previous findings on its colocalization at the DNA damage site along with other DDR factors outside of the telomeric regions have shed a light on the more extensive role it might play in regulating and maintaining DNA stability as well as its clinical implications. Some extra-telomeric interactions of TRF2 with DDR factors have been established as discussed in this review. However, very few mechanisms of such interactions have been determined and explored. The investigation on the molecular mechanism in the possible interaction between TRF2 and other DDR factors, either directly or indirectly, is needed to confirm whether there is a specific role TRF2 plays in DNA damage repair, if any exists. Additionally, as TRF2 protein level has been found to be elevated in some cancer cells and has an important role in regulating DDR, it is worth exploring if there is any possible therapeutic potential in targeting TRF2 or its known interacting factors.

## Figures and Tables

**Figure 1 ijms-22-09900-f001:**
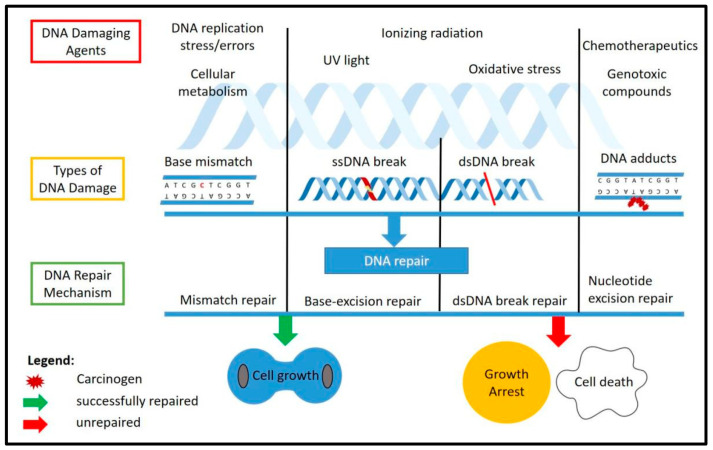
DNA damage and repair mechanisms. The figure illustrates the common DNA damaging agents (**top**), the types of DNA damage caused by these agents (**middle**) and the known response and repair mechanisms involved (**bottom**) as well as the possible outcomes after DNA repair, which include normal cell growth if the repair was successful, and growth arrest or programmed cell death if the damage was unable to be repaired. ss: single-stranded, dd: double-stranded.

**Figure 2 ijms-22-09900-f002:**
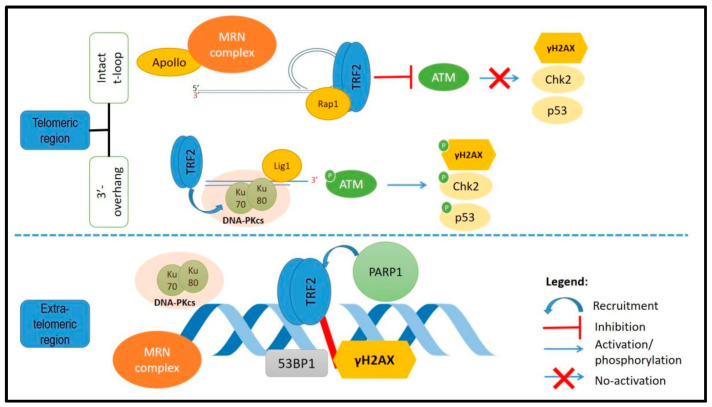
DNA damage response factors involved with TRF2 at the telomeres or outside of the telomeres. At the telomeres, TRF2 will maintain the t-loop structure and inhibit the ATM kinase pathway activation. Outside of the telomeres, TRF2 was reported to be recruited at the DNA damage site and colocalized with other DDR factors.

**Table 1 ijms-22-09900-t001:** Post-transcriptional modifications of TRF2.

Modification	Amino Acids	Modifying Enzymes	Function	Reference (s)
Phosphorylation	Ser20	Chk2	Phosphorylation of TRF2 by Chk2 decreases TRF2 binding to telomeric DNA. S20 phosphorylation is required for TRF2 interaction with G-rich RNA and recruitment of ORC to OriP.	[74,75]
	Ser365	CDK	Regulates the assembly and disassembly of t-loop during the cell-cycle or DNA replication, which protects the telomeres from replicative stress.	[69]
	Thr188	ATM	Early recruitment at the DNA break site and may be involved in DNA damage response/repair.	[72,95]
	Thr358	Aurora C	Possible function in cell replication and unwinding of the chromosome.	[96]
SUMOylation	IK140TE, LK245SE, and RK333DE	MMS21	Stabilization of TRF2 and recruitment at the telomeres.	[79,80]
	-	PIAS1	Interacts with RNF4 to regulate TRF2 level at the telomeres.	[78]
Ubiquitylation	K173 K180 K184	Siah1	Siah1 is a p53-inducible E3 ligase with a C3H4-type RING finger domain. Knockdown of Siah1 stabilizes TRF2 and telomere length maintenance.	[85]
	-	RNF4	Interacts together with PIAS1 in modulating TRF2 role in telomere maintenance and protection through regulating the level of TRF2 at the telomeres. Acts downstream of the ATM-kinase pathway in DDR.	[78,84]
Acetylation	K293	P300	Required for TRF2 stabilization and efficient binding at the telomeres.	[86]
Poly(ADP-ribosyl)ation or PARsylation	-	PARP1, PARP2	Reduces the binding of TRF2 to telomeres that may contribute to dysfunctional telomeres.	[92,97]

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
