# Peer review of "The Intra- and Extra-Telomeric Role of TRF2 in the DNA Damage Response"

_ijms, 2021, doi:10.3390/ijms22189900_

Round 1

Reviewer 1 Report

Overall the review is quite well written, however it is not complete and two important issues are only superficially or not all addressed. This is a major lack and has to be corrected.

First: The clinical impact of TRF2 deregulation (lines 82-92) is only superficially discussed.

The entire impact of TRF2 dislocation, up/down-regulation in the pathogenesis of Hodgkin’s lymphoma and the impact of EBV oncoprotein LMP1 expression on TRF2 down-regulation with following chromosomal rearrangements (BBF-cycles) has to be dealt with. The corresponding papers in top rated journals have to be addressed. (Leukemia 2009;23: 565-573; Blood 2015;125:2101-2110; Lab Invest 2017;97: 772-781).

Second: In 9. Other known extra-telomeric functions of TRF2 (line 336 and following)  to include: Lamin A/C is bound to to telomeres and expression of a doninant negative telomere repeat binding factor 2 (TRF2)  mutant leads to loss of telomere integrity and to the formation of DNA damage foci at telomeres.Lamin A/C is essential for maintaining the chromosomal nuclear territories and this occurs partially through direct interaction with TRF2 bound to t-loops at ITS, so-called interstitial t-loops (ITL). Impairment or loss of this 3D lamin A/C-TRF2 interaction at ITLs  results in increased chromatin dynamics, changes in gene expression, impaired chromosome stability and genomic integrity. (EMBO J2009; 28:2414-2427; Nat Commun2014;5:5467; Nat Commun 2015;6:804; Differentiation 2018;102:19-26: Nucleus 2015;6:172-178).

Once these issues are introduced in the text, small corresponding changes in Abstract and Conclusions & Future Directions have to be arranged.  

Author Response

Please find the reply as attached.

Reviewer 2 Report

The manuscript entitled ‘The intra- and extra-telomeric role of TRF2 in the DNA Damage Response’ by Siti A. M. Imran et al represents a very interesting and comprehensive review manuscript where the authors highlighted the complexity of intra- and extra telomeric role TRF2 in DNA damage response. Further, the role of TRF2 in DNA damage repair mechanisms was also explored and discussed. The original figures and tables are quite useful.

In overall, I consider that the premise of this study is very interesting and important for the field, and I will perform some comments and suggestions.

Minor concerns:

  1. In figure 1, there is no correspondence between the DNA damaging agents and the type of DNA damage observed. Is it possible to put the agents in the way that the readers can correlate the agent used/type of DNA damage and mechanism of repair?
  2. In the text S20 TRF2 phosphorylation is not mentioned.
  3. The Title ‘5. TRF2 inhibits the ATM Kinase’ is not formatted (line 219).
  4. In Table 1-SUMOylation referent to amino acids of PIAS1 is not completed.
  5. In Table 2, LAP1, ‘A novel helicase that was recently found to be involved in DDR’. Is LAP1 considered a helicase???
  6. No supplementary material was presented, and that section should be removed (line 393-394)
  7. At introduction, confirm the repetitive use of the following expressions:‘has been shown to be crucial’ (line 36); ‘has been shown to be important’ (line 38); ‘has been shown to be colocalized’ (line 43)

Author Response

Dear Reviewer, 

Please find the response as attached.

Round 2

Reviewer 1 Report

Overall, the two major lacking pieces of important advances, i.e. the two major criticisms have been adequately answered and the review is now quite complete and offers a nice overview about the central role of the shelterin protein TRF2 in DNA damage and repair response in the setting of translational research and corresponding implications on disease progression.

There are still a few spelling errors identifiable.